# A squeezed quantum microcomb on a chip

Zijiao Yang[1,2,5], Mandana Jahanbozorgi[1,5], Dongin Jeong[3], Shuman Sun[1], Olivier Pfister[2], Hansuek Lee [3,4] & Xu Yi [1,2✉]

The optical microresonator-based frequency comb (microcomb) provides a versatile platform for nonlinear physics studies and has wide applications ranging from metrology to spectroscopy. The deterministic quantum regime is an unexplored aspect of microcombs, in which unconditional entanglements among hundreds of equidistant frequency modes can serve as critical ingredients to scalable universal quantum computing and quantum networking. Here, we demonstrate a deterministic quantum microcomb in a silica microresonator on a silicon chip. 40 continuous-variable quantum modes, in the form of 20 simultaneously two-mode squeezed comb pairs, are observed within 1 THz optical span at telecommunication wavelengths. A maximum raw squeezing of 1.6 dB is attained. A high-resolution spectroscopy measurement is developed to characterize the frequency equidistance of quantum microcombs. Our demonstration offers the possibility to leverage deterministically generated, frequency multiplexed quantum states and integrated photonics to open up new avenues in fields of spectroscopy, quantum metrology, and scalable, continuous-variable-based quantum information processing.

[1] Department of Electrical and Computer Engineering, University of Virginia, Charlottesville, VA, USA. [2] Department of Physics, University of Virginia, Charlottesville, VA, USA. [3] Graduate School of Nanoscience and Technology, Korea Advanced Institute of Science and Technology (KAIST), Daejeon, South Korea. [4] Department of Physics, Korea Advanced Institute of Science and Technology (KAIST), Daejeon, South Korea. [5]These authors contributed equally: Zijiao Yang, Mandana Jahanbozorgi. ✉email: yi@virginia.edu

Optical microresonators employ the Kerr nonlinearity[1] to provide broadband parametric gain through four-wave mixing among cavity resonance modes, where pairs of pump photons can be annihilated to generate signal and idler photons at lower and higher frequencies. The Kerr parametric process has been used to demonstrate microresonator-based frequency combs (microcombs)[2,3] and dissipative Kerr cavity solitons[4–7], which have revolutionized a wide range of applications from metrology[8] to spectroscopy[9]. The quantum aspects of microcomb have been studied recently[10–15] for its capability of providing hundreds of frequency multiplexed quantum channels from a single microresonator. Access to individual quantum channels is possible through off-the-shelf wavelength-division-multiplexing filters thanks to microcombs' large free-spectral-ranges (FSRs), which range from a few GHz to 1 THz[16,17] as opposed to the finer FSRs of fiber or bulk resonator-based combs. When combined with integrated photonic circuits, quantum microcombs have the potential to revolutionize photonic quantum information processing.

So far, experiments of quantum microcombs have been limited to the probabilistic regime[11–14], where entanglement is measured between randomly emitted photon pairs with postselecting, coincidence detection. The photon coincidence rate suffers from exponential decrease with the increase of photon number in a quantum state. Quantum architectures built upon probabilistic quantum states are not scalable without quantum memory, which allows repeat-until-success strategies[18,19]. In contrast, a quantum microcomb in the deterministic regime, where the entanglement among different frequency modes can be deterministically generated and detected, will be a significant step forward towards the scalable quantum architecture on photonic chips.

One approach to constructing deterministic quantum microcombs is to leverage two-mode squeezing and create unconditional entanglement between the optical fields in optical frequency combs[20–22]. Squeezed light[23], with quantum uncertainty below than that of the vacuum field, has broad applications in science and technology, ranging from enhancing the gravitational wave detection sensitivity in LIGO[24], Gaussian boson sampling[25,26], to continuous-variable-based quantum computing (CVQC)[27–30]. The unconditional entanglement created by two-mode squeezing is between continuous optical fields, which can serve as quantum modes (qumodes) to encode quantum information through continuous-variable-based (CV) approaches[31] for applications in universal quantum computing[27,28,30], unconditional quantum teleportation[32], quantum dense coding[33], quantum secret sharing[34], and quantum key distribution[35]. Unlike probabilistic photonic qubit approaches, the unconditional entanglement in CV approaches enables the number of entangled quantum modes (qumodes) in a quantum state to be deterministically scaled up through frequency[20,21,36], time[37–39], or spatial multiplexing[40], which provides a scalable physical platform for continuous-variable quantum computing[31]. Squeezing is conventionally generated through nonlinear optics in bulk optical systems, such as optical parametric oscillators (OPOs)[23,41], or atomic vapor[42,43]. Squeezed quantum microcombs, when combined with integrated photonic circuits, Gaussian and non-Gaussian measurements, can serve as simple and compact building bricks for CV universal quantum computing[44], entanglement-assisted spectroscopy[45], and quantum networking for distributed quantum sensing[46]. While the generation[47–56] and detection[57] of one or two squeezed frequency qumodes, and the generation of 8 spatial qumodes[26] have been shown in miniaturized platforms recently, a squeezed microcomb has not been reported yet.

In this work, we demonstrate a deterministic, two-mode-squeezed quantum frequency comb in a silica microresonator on a silicon chip. The Kerr parametric process generates unconditional Einstein–Podolsky–Rosen entanglement, i.e., two-mode squeezing, between the optical quadrature fields of the qumode pairs in the microresonator[10,55]. 40 frequency multiplexed qumodes, in the form of 20 two-mode squeezed comb pairs, are measured at telecommunication wavelengths. The two-mode squeezing is verified by measuring the noise variance through balanced homodyne detection. The maximum raw squeezing of 1.6 dB and maximum anti-squeezing of 6.5 dB are attained. A corresponding 3.1 dB squeezing at the output waveguide can be inferred after correcting system losses. A qumode spectroscopy measurement is developed to characterize the frequencies of qumodes with a resolution of 5 MHz in 1 THz optical span. The frequency equidistance of qumodes can simplify homodyne detection in CV quantum information processing, as a simple laser frequency comb can serve as local oscillators (LOs) for all qumodes. In our experiments, the number of accessible qumodes is limited by the a 1 THz optical span of the LOs. The optical span of a quantum microcomb is ultimately set by the chromatic dispersion of the microresonator, which determines the bandwidth of Kerr parametric gain.

## Results

**Squeezed quantum microcombs**. The microresonator used in this work is a 3 mm diameter silica wedge resonator with 22 GHz FSR on a silicon chip[58] (Fig. 1). The resonator's intrinsic quality factor ($Q_o$) is 79 million, and a single-mode tapered fiber is used as the coupling waveguide. The resonator is overcoupled to achieve large field escape efficiency, $\eta = \kappa_c/(\kappa_c + \kappa_0)$, of 83%, where $\kappa_0$ and $\kappa_c$ are the intracavity dissipation rate and the resonator-waveguide coupling rate. An amplified continuous-wave (cw) laser is used as the pump of the resonator, and its frequency ($\omega_p$) is phase-locked to the resonance mode at 1550.5 nm through Pound–Drever–Hall (PDH) locking technique[59]. The pump power in this experiment is set to 120 mW, which is 0.5 dB below the parametric threshold of 135 mW. The dependence of squeezing on the input pump power is discussed in the Methods section. At the through port of the resonator, a narrow-band fiber-Bragg grating (FBG) filter is used to filter out the pump field and the amplified spontaneous emission (ASE) noise from the erbium-doped fiber amplifier (EDFA). In principle, the bichromatic LOs for two-tone homodyne detection can be derived from a soliton microcomb[6]. However, in the present measurement, an electro-optic modulation (EOM) frequency comb[60] is used, where tens of modulation sidebands are created by strong electro-optic phase modulation at modulation frequency $f_m$ on the cw-laser. A programmable line-by-line waveshaper is then used to select a pair of comb lines as the LOs (see Fig. 2). A periodic ramp voltage is applied on a phase modulator (PM) with $V_\pi = 2.3$ V to scan the phase of the LOs. The LOs and the resonator pump laser are coherent with each other since they are derived from the same cw-laser, and the electro-optic modulators coherently transfer photons from the pump to the modulation sidebands. A detailed description of the experimental setup is provided in the "Methods" section.

The quadrature noise variances of 20 sets of comb pairs (40 qumodes) are measured by means of balanced homodyne detection. To measure the quadrature noise variance of qumodes $(-N, N)$, the EOM comb modulation frequency $f_m$ and the programmable waveshaper are adjusted to precisely match the frequencies of LO pairs to $\omega_p \pm N \times D_1$, where $N$ is the relative mode number from the mode being pump ($N = 0$), and $D_1/2\pi = 21.95258$ GHz is the FSR of the resonator at 1550.5 nm wavelength. In each measurement, the phase of the LOs is ramped to yield varying quadrature variances. Figure 2b shows a

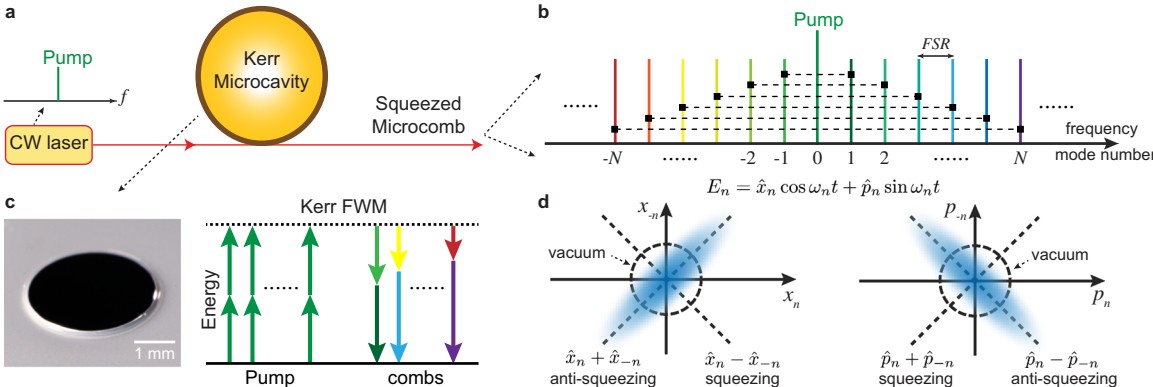

**Fig. 1 Generation of deterministic, two-mode squeezed quantum microcombs on a chip. a** A continuous-wave (cw) pump laser is coupled to a microresonator, which has thousands of longitude resonance modes with their frequencies separated by the resonator free-spectral-range (FSR), as shown in (**b**). **c** The $\chi^{(3)}$ Kerr nonlinearity in the microresonator creates broadband parametric gain as the pump photon pairs (green) can be converted into signal and idler photons at lower and higher frequency modes. This nonclassical correlation creates two-mode vacuum squeezing and thus unconditional EPR entanglement of the optical quadrature fields between frequency modes $n$ and $-n$, which are connected by dashed black lines in the optical spectrum in (**b**). Also shown is the image of a silica microresonator on a silicon chip used in this work. **d** Conceptual illustration of the two-mode squeezing wavefunctions in position (left) and momentum (right) basis, where $(\hat{x}_n - \hat{x}_{-n})$ and $(\hat{p}_n + \hat{p}_{-n})$ have uncertainty level below the vacuum fluctuation (dashed circle). The electrical field of the $n$-th optical mode is $E_n = \hat{x}_n \cos\omega_n t + \hat{p}_n \sin\omega_n t$, where $\hat{x}_n$ and $\hat{p}_n$ are the in-phase and out-of-phase quadrature amplitudes of the mode at frequency $\omega_n$.

representative quadrature noise variance (blue) relative to the shot noise (red) for qumodes (−4, 4). A 30-point moving average is used to smooth out the fluctuations in the noise variance measurement. The raw squeezing of $1.6 \pm 0.2$ dB and anti-squeezing of $5.5 \pm 0.1$ dB are directly observed, and they are obtained by averaging the displayed extrema. The uncertainty is concluded with a 95% confidence interval under t-distribution. The quadrature noise variances of all 40 qumodes are shown in Fig. 2c, and squeezing/anti-squeezing are observed for all 40 qumodes. The number of measurable qumodes is limited by the 1 THz optical span of the EOM comb. All measurements are taken at 2.7 MHz frequency, 100 kHz resolution bandwidth (RBW), and 100 Hz video bandwidth (VBW) on an electrical spectrum analyzer (ESA). The noise levels of qumodes (−1, 1) to (−3, 3) are not presented here as their measurements are affected by the transmitted ASE noise from the EDFA near the pump frequency. This can be addressed in the future by using a filter with bandwidth much smaller than the FSR of the resonator, or by increasing the intrinsic quality factor of the cavity and reducing the parametric oscillation threshold to eliminate the need for the EDFA. Finally, as shown in Fig. 2d, no quantum correlation (two-mode squeezing) is observed for uncorrelated comb pairs. This serves as a critical check for our two-tone homodyne detection.

The raw squeezing and anti-squeezing levels of all 40 qumodes are summarized in Fig. 3a. The raw squeezing in our experiment is primarily limited by the 83% cavity escape efficiency, 1.7 dB optical loss, and ~89% photodiode quantum efficiency. The total efficiency after the tapered fiber is 60%. Our $1.6 \pm 0.2$ dB raw squeezing is among the highest raw squeezing measured for miniaturized Kerr OPOs[56], while the highest squeezing ever achieved is 15 dB in a bulk $\chi^{(2)}$ OPO[61]. 6 dB single-mode squeezing was reported earlier in an integrated waveguide[53], which indicates that large squeezing is possible in integrated photonic platforms. Recent theoretical studies have suggested that quantum error correction and fault-tolerant quantum computing is possible in photonic CV-based approaches[30] when squeezing reaches 10 dB[62].

The anti-squeezing levels near qumodes (−10, 10), and from (−17, 17) to (−23, 23) are observed to be smaller than that of other qumodes. We suspect this is caused by spatial-mode

interaction between different transverse mode families in the resonator, which not only modifies local dispersion[63] but provides a path to dissipate optical fields from the squeeze-generating mode to another spatial mode[64]. The spatial-mode-interaction can be identified by measuring the frequency spectrum of a resonator. The relative mode frequencies of the resonator, $\Delta\omega_N = \omega_N - \omega_0 - N \times D_1$, are measured with sideband spectroscopy method[65] and presented in Fig. 3b, where $\omega_N$ is the resonance frequency of relative mode number $N$. An avoided mode crossing[63] was found near mode −8, and resonance frequencies below mode −18 and above mode 19 are observed to change abruptly. These are likely caused by the spatial-mode interaction and hybridization between two transverse cavity modes. The mode numbers that are affected by spatial-mode-interaction in the mode spectrum coarsely align with that of the dips in anti-squeezing measurement. More systematic studies will be performed in the future to understand the mechanism of how spatial-mode-interaction affects squeezing and anti-squeezing. Finally, the impact of spatial-mode interaction can be eliminated in the future by using a microresonator with a single transverse mode family[66,67].

**Squeezed microcomb qumode spectroscopy**. A qumode spectroscopy method is developed to characterize the frequency equidistance of squeezed qumodes, a prerequisite of frequency combs. Similar to the classical cavity mode spectrum, we can define the relative qumode spectrum as $\Delta\omega_N^Q = \omega_N^Q - \omega_0 - N \times D_1$, where $\omega_N^Q$ is the optical frequency center of the $N$-th qumode. The relative qumode spectrum represents the qumode frequency deviation from equidistance. To identify the relative qumode spectrum, the two-sided squeezing/anti-squeezing spectral line shape is measured for each pair of qumodes, and the center frequency of the spectral line shape yields the relative qumode frequency. In the measurement, the $\pm N$-th LO frequencies are detuned by $\pm\delta$ from the equidistant frequencies, $\pm N \times$ FSR, and noise variances are measured at each detuning point for qumodes (−$N$, $N$). For each pair of qumodes, the detuning ($\delta$) is varied from −30 MHz to +30 MHz with an interval of 5 MHz, which sets the resolution of the line shape measurement. Measurements of qumodes (−4, 4) at $\delta = -20, -10, 0, 10, 20$ MHz

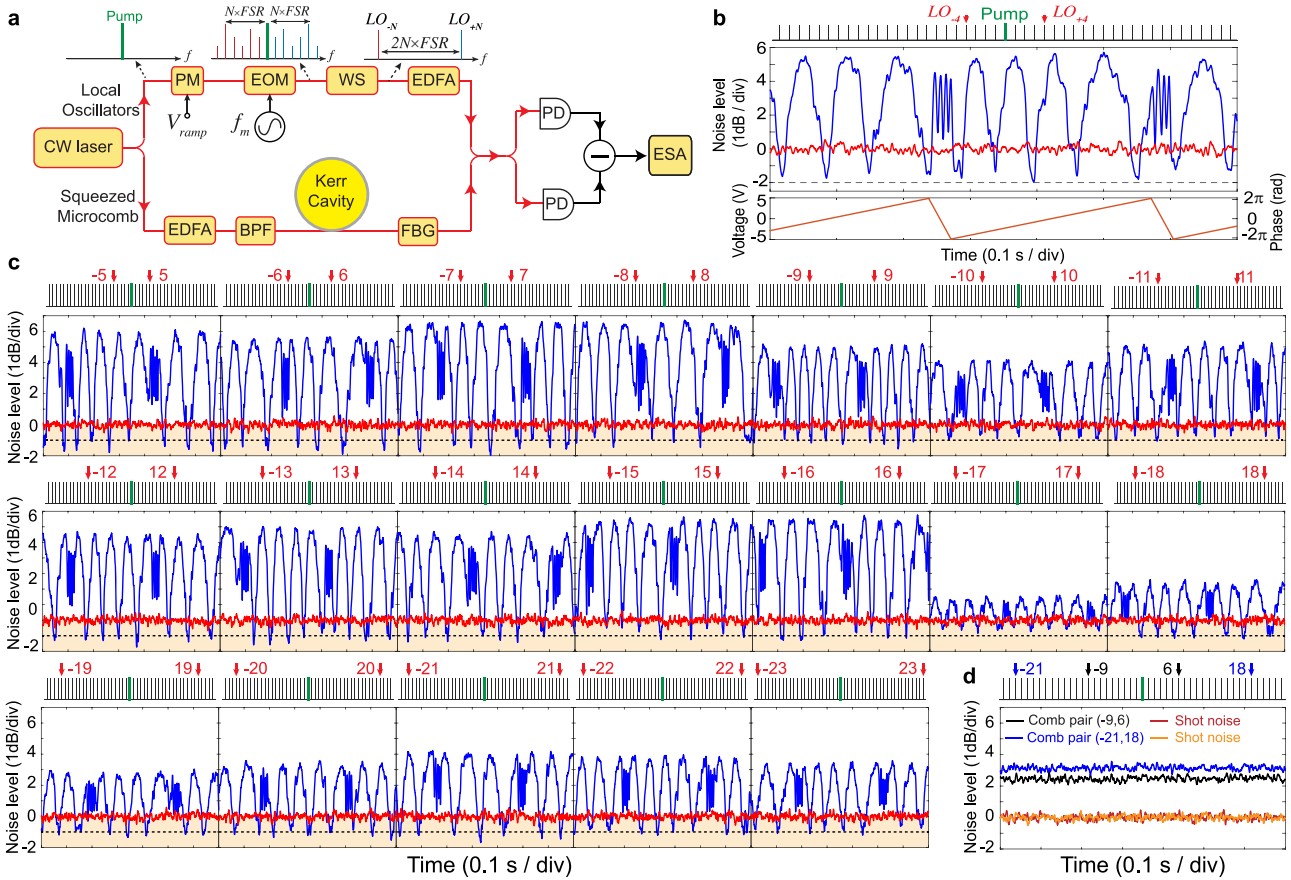

**Fig. 2 Two-mode squeezing measurement of 20 comb pairs (40 qumodes) from the microresonator. a** Simplified experimental schematic. A continuous-wave (cw) laser is split to pump the resonator and drive the local oscillators (LOs). The LOs are derived from an electro-optic modulation (EOM) frequency comb, with a comb spacing (modulation frequency) of $f_m$. A line-by-line waveshaper (WS) is used to select a pair of comb lines as the bichromatic local oscillators. The phase of the LOs can be tuned by a phase modulator (PM). The LOs and the squeezed microcombs are combined by a 50/50 coupler and are detected on balanced photodiodes (PDs). The noise level is characterized on an electrical spectrum analyzer (ESA). In the squeezed microcomb path, a fiber-Bragg grating (FBG) filter is used to block the strong pump light. Erbium-doped fiber amplifiers (EDFAs) and optical bandpass filter (BPF) are also shown in the figure. **b** Representative quadrature noise variance (blue) relative to shot noise (red) as a function of time for qumodes −4 and 4 (indicated with red arrows). The lower panel illustrates the ramp waveform applied to the phase modulator to ramp the phase of the LOs periodically with time. 1.6 dB squeezing and 5.5 dB anti-squeezing are directly observed. A dashed black line indicates 2 dB below shot noise level. **c** Quadrature noise variances (blue) relative to shot noise (red) of all 40 qumodes. The qumodes measured are marked by the red arrows. The regime below the shot noise limit is colored in orange, and a dashed black line indicates 1 dB below the shot noise level. **d** Quantum correlation check: noise variances show no quantum correlation between uncorrelated comb pairs for qumodes (−9, 6) and (−21, 18). All measurements are taken at 2.7 MHz frequency, 100 kHz resolution bandwidth, and 100 Hz video bandwidth.

are shown as examples in Fig. 4b. At each detuning point, squeezing and anti-squeezing levels can be extracted by averaging the extrema. We plot the squeezing/anti-squeezing levels versus detuning ($\delta$) for all qumodes in Fig. 4c, which manifest the two-sided spectral line shape of the qumodes. The squeezing/anti-squeezing extraction below 0.5 dB has relatively poor accuracy, but this does not affect the overall qumode spectrum envelopes.

The relative frequencies of the qumodes, i.e., relative qumode spectrum, can be obtained by identifying the centers of the anti-squeezing line shapes via Lorentzian fitting. The average root mean square deviation of the fitting is only 0.15 dB, showing an excellent agreement between fitting and measurements. $\Delta\omega_N^Q$ of all the qumodes are plotted in Fig. 4d, and their deviations from equidistant are within the 5 MHz spectroscopy resolution limit for the entire 1 THz optical span of the quantum microcomb. The qumode spectrum overlaps well with the two-sided averaged cold cavity mode spectrum, $-(\Delta\omega_N + \Delta\omega_{-N})/2$, which represents the averaged deviation from equidistant of the cold cavity mode $N$ and $-N$. It should be noted that in the qumode spectrum

measurement, the cavity is pumped by >100 mW power, which could alter the cavity mode spectrum through thermo-optic effect and self/cross-phase modulation effects. Further study in the future is necessary to understand the requirement for perfectly equidistant frequencies of qumodes. In this measurement, the cavity escape efficiency is adjusted to 77% to achieve a more stable coupling condition as the entire measurement spans over 18 hours. As a result, the amount of squeezing/anti-squeezing at $\delta = 0$ MHz is different from that in the Fig. 2. In this experiment, the escape efficiency is adjusted by varying the relative position between the microresonator and the tapered fiber[68]. The stability of the escape efficiency can be dramatically improved by packaging the microresonator systems[69], or by integrating the coupling waveguide and the resonator on the same chip[67].

## Discussion

In summary, we have demonstrated a deterministic two-mode-squeezed quantum microcomb in a silica microresonator on a

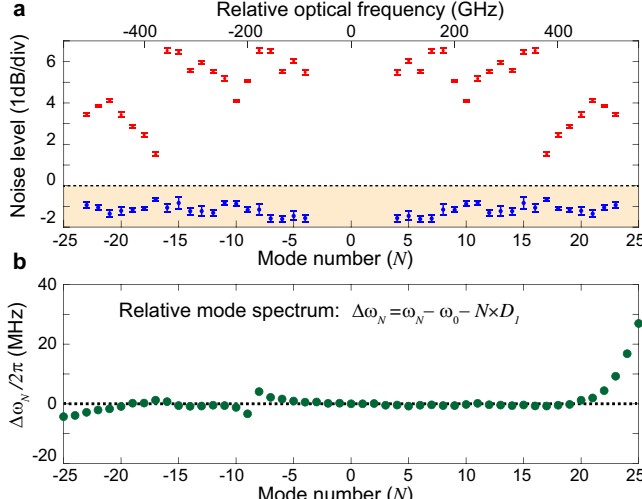

**Fig. 3 Summary of squeezing and anti-squeezing levels and resonator mode spectrum. a** Squeezing (blue) and anti-squeezing (red) levels versus mode number. The regime below the shot noise level is colored in orange. It should be noted that the noise level at qumode $N$ or $-N$ represents the two-mode noise level of comb pair $(-N, N)$. The error bars are concluded with a 95% confidence interval under t-distribution. **b** The cold resonator mode spectrum ($\Delta\omega_N$, relative mode frequency). The degradation of squeezing/anti-squeezing level of certain qumodes is likely caused by the avoided mode crossing induced by spatial-mode-interaction in the microresonator.

silicon chip. The generation of squeezed microcomb is not limited to Kerr microresonators, but can also be realized in microresonator-based $\chi^{(2)}$ parametric oscillators[70–73]. For our system, the raw squeezing can be improved in the future by reducing system losses, improving photodiode quantum efficiency, and achieving higher resonator-waveguide escape efficiency. The number of measurable qumodes, 40, is primarily limited by the span of the local oscillator, and this could be dramatically increased in the future by spectrum broadening of the EOM comb[60], or by using broadband dissipative Kerr soliton microcombs[6] as the LOs. The optical span of quantum microcombs will ultimately be limited by the microresonator dispersion, which sets the bandwidth of Kerr parametric gain. Through dispersion engineering, Kerr parametric sidebands that are ~ ±80 THz away from the pump frequency have been reported in microresonators[74], which indicates the possibility of creating hundreds or thousands of qumodes in a single microresonator. The miniaturization of deterministic quantum frequency combs provides a path towards mass production, which could be critical for applications in quantum computing, quantum metrology, and quantum sensing[75,76].

## Methods

### Experimental setup.
The experimental setup is shown in Fig. 5a. A continuous-wave (cw) laser (New Focus, TLB-6700) at 1550.5 nm is used to drive both the squeezed microcomb and the LOs. For the squeezed microcomb generation, the cw laser is amplified by an EDFA to pump the Kerr microresonator. A FBG filter is used to filter out the ASE noise from the EDFA. The amplified pump laser is then coupled into the microresonator through a single-mode tapered fiber. At the through port of the tapered fiber, another FBG filter is used to separate light at the pump laser wavelength from light at all other wavelengths. The transmitted squeezed microcomb from the FBG is sent to a 50/50 fiber coupler to be combined with the LOs for balanced homodyne detection. In the experiment, the PDH locking technique is used to lock the pump laser frequency to the resonator mode frequency by servo control of the cw-laser frequency. This is implemented by phase

modulating the pump laser before the EDFA with an electro-optic PM, and then photodetecting the pump laser after the second FBG filter. The phase modulation frequency is set to 80 MHz, much higher than the resonator linewidth. It should be noted that the Brillouin scattering does not affect the squeezing process in our resonator, as the resonator FSR is designed to be completely out of the Brillouin gain bandwidth[58]. Raman scattering in silica has its peak gain at 13 THz away from the pump, and the peak Raman gain is smaller than the Kerr parametric gain in microresonators with anomalous dispersion[1]. As the optical span of the quantum microcomb is only ±0.5 THz around the pump, the Raman gain within our microcomb span is only ~2% of the peak Raman gain, and it has a negligible effect in our current experiment. The effect of Raman scattering on wideband quantum microcombs will be studied in the future.

### Electro-optic modulation frequency comb.
The LOs in this experiment are derived from an EOM frequency comb[77]. The EOM comb is convenient to create coherent LOs which are hundreds of GHz apart from the pump laser frequency. In our EOM comb, the cw laser is amplified by an EDFA to 200 mW and is phase modulated by three cascaded electro-optic PMs at frequency $f_m$, which is provided by a signal generator (Keysight, PSG E8257D). The modulators are driven by amplified electrical signals that are synchronized by electrical phase shifters (PSs). The output power of the electrical amplifiers (Amps) is ~33 dBm. As the EOM comb and the microresonator share the same pump laser, the LOs derived from the EOM comb are inherently coherent with the squeezed microcomb. A typical EOM comb spectrum is shown in Fig. 5b (blue line), and the cw pump laser spectrum (black) is also shown as a reference. The EOM comb is then sent to a programmable line-by-line waveshaper (Finisar 1000A, filter bandwidth setting resolution: ±5 GHz), which can control the amplitude and phase of each EOM comb line. To measure the noise variance of qumodes $(-N, N)$, the waveshaper is set to only pass the comb lines whose frequencies are $\pm N \times$ FSR apart from the pump laser. As an example, the LOs for qumodes $(-21, 21)$ are shown in Fig. 5b (red line). Finally, the LOs are amplified to ~17 mW and are combined with the squeezed microcomb for balanced homodyne detection. It should be noted that the relative phase between the local oscillator and the squeezed field could be different from the phase shift applied by the PM in the LO optical path. This is because environmental fluctuations, e.g., ambient temperature, can cause phase variations in fibers in both LO and squeezed light paths. Finally, the electrical amplifiers in the EOM comb cutoff at 18 GHz, which is smaller than the FSR of the resonator (represented by $f_r$). As a result, the EOM modulation frequency, $f_m$, is set to $n/m \times f_r$, such that the frequency of the $m$-th EOM comb line can align with that of the $n$-th resonator mode. The modulation frequencies used in the experiment are: $f_m = 3/4 \times f_r = 16.464438$ GHz for mode pairs $\pm 6$, $\pm 9$, $\pm 12$, $\pm 15$, $\pm 18$, and $\pm 21$; $f_m = 2/3 \times f_r = 14.635056$ GHz for mode pairs $\pm 4$, $\pm 8$, $\pm 14$, $\pm 16$, and $\pm 20$. For mode pairs $\pm 5$, $\pm 7$, $\pm 11$, $\pm 13$, $\pm 17$, $\pm 19$, $\pm 22$, and $\pm 23$, modulation frequencies of: $5/7 \times f_r = 15.680417$ GHz, $7/9 \times f_r = 17.074232$ GHz, $11/15 \times f_r = 16.098562$ GHz, $13/17 \times f_r = 16.787270$ GHz, $17/23 \times f_r = 16.225823$ GHz, $19/25 \times f_r = 16.683964$ GHz, $22/29 \times f_r = 16.653684$ GHz, and $23/29 \times f_r = 17.410670$ GHz are used, respectively.

The phase noise of the signal generator that drives the EOM comb contributes to the phase fluctuation of the local oscillator, which could potentially affect squeezing measurement[78]. Here, we estimate its impact on our experiments. The root mean square (RMS) of phase jitter from the signal generator can be calculated from its single-sideband phase noise by integrating the phase noise from the electrical spectrum analyzer (ESA) VBW used in the squeezing measurement (100 Hz), to the bandwidth of our balanced photodetection circuit (250 MHz). The RMS of phase jitter ($\tilde{\theta}$) is calculated to be 0.0024 rad (0.14°) for comb mode 1 (~22 GHz), and is 0.055 rad (3.2°) for comb mode 23 (~0.5 THz). After taking account of this phase fluctuation, the observable level of squeezing[78] is $R' \approx R_S \cos^2\tilde{\theta} + R_{AS}\sin^2\tilde{\theta}$, where $R_S$ and $R_{AS}$ are the variance of output squeezing and anti-squeezing, respectively. For the current experimental condition, assuming 2 (7) dB squeezing (anti-squeezing) at mode 4, and 1 (5) dB squeezing (anti-squeezing) at mode 23 after optical losses, the phase fluctuation will cause the measured squeezing ($R'$) to be 0.003 and 0.04 dB lower than the actual squeezing ($R_S$) at mode 4 and mode 23, respectively. It should be noted that the $N$-th comb line in the EOM comb has $N$ times the RMS phase jitter of the 1st comb line in the EOM comb. Therefore, when scaling up the number of comb lines in an EOM comb through supercontinuum generation[60] for squeezing measurement, the phase noise of the signal generator should be improved accordingly to maintain the low phase fluctuation of the LOs. A possible way to obtain exceptional phase noise performance for the EOM comb is through electro-optical frequency division, where the signal generator is synchronized to stable optical references[79].

### Characterization of balanced photodiodes.
In the two-mode squeezing noise variance measurement, the balanced photodiodes (JDSU, ETX 300T) are operated in the shot noise-limited regime. The electrical circuit for balancing the photodiodes is home-built[21], and a common-mode rejection ratio of 31 dB is measured. The shot noise-limited regime is verified by the linear relationship

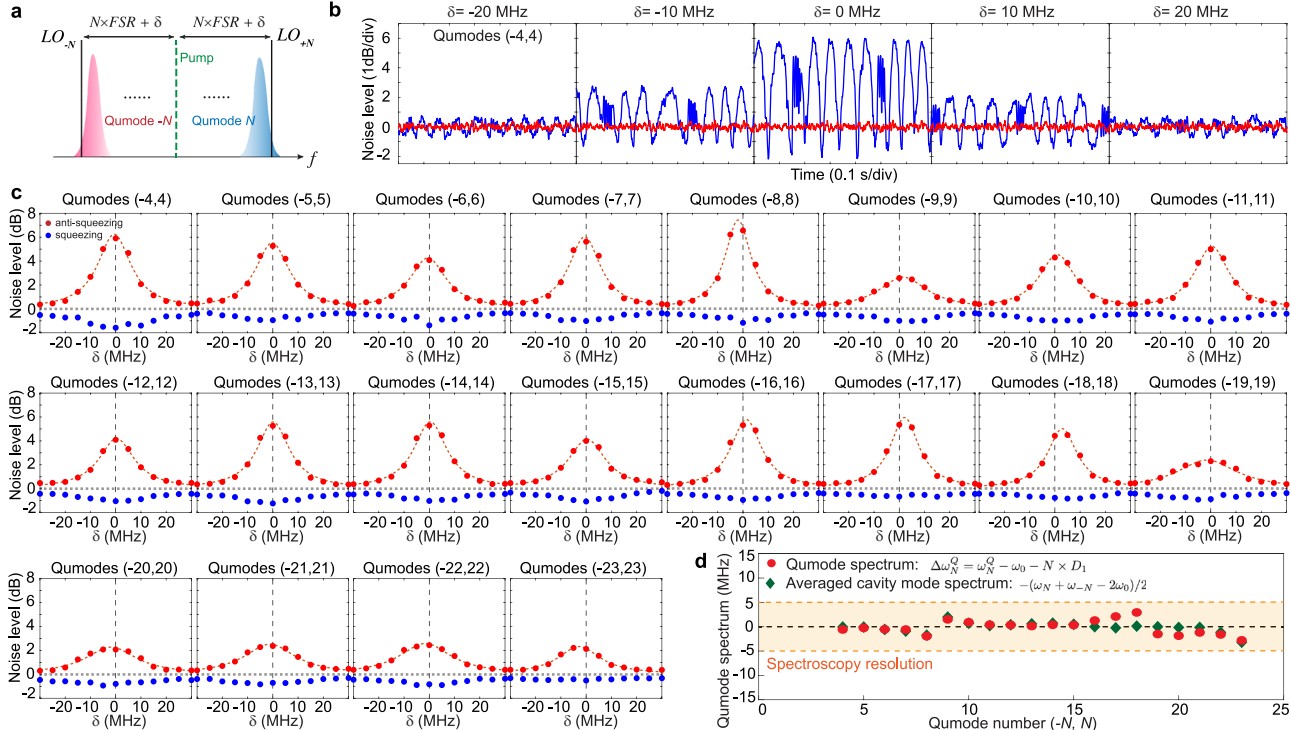

**Fig. 4 Spectroscopy characterization of qumodes in the squeezed quantum microcomb. a** Illustration of spectroscopy measurement of qumode $(-N, N)$. The frequencies of the $N$-th LOs can be detuned by $\delta$ away from the equidistant frequencies, $\pm N \times$ FSR, and the amount of squeezing and anti-squeezing are measured at each detuning point, $\delta$. **b** Noise variance measurement of qumodes $(-4, 4)$ at detuning $\delta = -20, -10, 0, 10, 20$ MHz. The red trace represents shot noise level. **c** Squeezing (blue) and anti-squeezing (red) levels extracted from noise variance measurements at different detuning points ($\delta$) for all qumodes. Shot noise levels are represented by the horizontal dashed gray lines. Lorentzian fitting of the anti-squeezing spectrum (red dash line) is used to find the qumode center frequencies. Vertical dashed black lines represent the equidistant frequencies for each qumodes. **d** Summary of the measured relative qumode frequencies (red) from qumodes $(-3, 3)$ to $(-23, 23)$. The two-sided averaged cavity mode spectrum: $-(\Delta\omega_N + \Delta\omega_{-N})/2$ is plotted in green and it agrees well with the qumode spectrum.

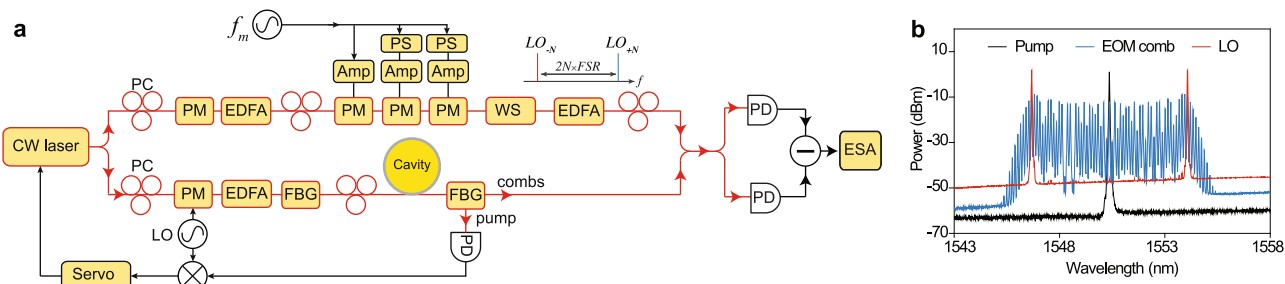

**Fig. 5 Experimental setup and optical spectrum of local oscillators. a** Optical and microwave components are colored in red and black boxes, respectively. A continuous-wave (cw) laser drives both the squeezed microcomb and the local oscillators. Part of the cw laser is amplified by an erbium-doped fiber amplifier (EDFA) to pump the silica microresonator. A fiber-Bragg grating (FBG) filter is used at the microresonator through port to separate the pump light and the squeezed light. The local oscillators are derived from an electro-optic modulation (EOM) frequency comb, which is driven by the same cw laser. The cw laser is phase modulated by three tandem phase modulators (PMs) at frequency $f_m$. A programmable waveshaper (WS) is used to select a pair of comb lines to be the local oscillators. The LOs and the squeezed microcomb are combined and detected on the balanced photodetectors (PDs). The noise variance is characterized by an electrical spectrum analyzer (ESA). Polarization controller (PC), electrical amplifier (Amp), and phase shifter (PS) are also included in this figure. **b** Optical spectra of the pump laser (black), the EOM frequency comb (blue), and the local oscillators (red) for qumodes $(-21, 21)$.

between the noise power of the balanced photodiodes and the optical input power, which is shown in Fig. 6a. The measurement is done at 2.7 MHz with 100 kHz RBW. The electrical spectra from the balanced photodiodes at different optical input powers are shown in Fig. 6b. The resonance peaks in the dark noise are likely caused by the electrical circuits in the balanced photodiodes. At 16.6 mW input power, the electrical spectrum is relatively flat. The spectra roll-off is around 20 MHz.

**Dependence of squeezing on optical pump power**. The dependence of squeezing and anti-squeezing on optical pump power is measured for qumode $(-4, 4)$ and is presented in Fig. 6c. Ideally, when there is no optical loss, vacuum squeezing should increase with the pump power until the pump power reaches the OPO threshold. However, as the amount of squeezing in our experiment is primarily limited by optical losses, the increase of squeezing can no longer be observed when the pump power is roughly above half of the OPO threshold. On the other hand, the anti-

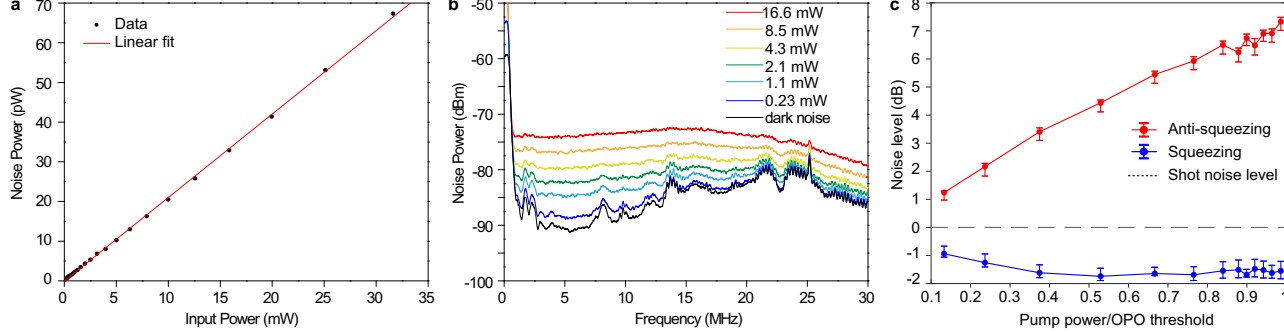

**Fig. 6 Characterization of the balanced photodiodes, and squeezing dependence on optical pump power. a** Noise power vs. the optical power of local oscillators sent into the PDs. The noise power is measured at 2.7 MHz frequency, and the dark noise from the PDs has been subtracted from the noise power. The linear trend indicates the balanced photodiodes are operated in the shot noise-limited regime. **b** Electrical spectra of the balanced PD outputs at different local oscillator powers. All measurements in this figure are taken at 100 kHz resolution bandwidth. **c** Measurement of squeezing and anti-squeezing versus pump power for qumode (−4, 4). The error bars are concluded with a 95% confidence interval under t-distribution.

squeezing increases with the pump power. This observation is consistent with measurements in previous reports[55].

## Data availability

Source data for Figs. 2–6 can be accessed at https://doi.org/10.6084/m9.figshare.14921670. Additional information is available from the corresponding author upon reasonable request.

## Code availability

The codes that support the findings of this study are available from the corresponding authors upon reasonable request.

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

## Acknowledgements

The authors thank A. Beling at UVA for the access of signal generator, Y. Shen and J. Campbell at UVA for assisting photodiode quantum efficiency calibration, and gratefully acknowledge National Science Foundation.

## Author contributions

X.Y., O.P., Z.Y., and M.J. conceived the idea and designed the experiments. Z.Y. and M.J. performed the measurements. D.J. and H.L. fabricated the microresonator. X.Y., O.P., Z.Y. M.J., and S.S. analyzed the data. All authors participated in preparing the paper and contributed to the discussions. X.Y. supervised the project.

## Competing interests

XXX.
