## [Peer Review File · Nature Communications]

REVIEWER COMMENTS

Reviewer #1 (Remarks to the Author):

Review of the manuscript "A squeezed quantum microcomb on a chip"

In the manuscript titled "A squeezed quantum microcomb on a chip" by Z. Yang et al., the authors demonstrate the realization of a deterministic quantum microcomb using an integrated silica micro-resonator. They use the microcomb as a platform for the generation of 40 continuous-variable (CV) quantum modes (qumodes) analyzed as 20 two-mode squeezed resonance pairs, emitted as signal and idler signals (generated through four-wave mixing within the micro resonator) spanning a 1 THz optical band at telecommunication wavelengths. The unconditional Einstein-Podolsky-Rosen (EPR) entanglement of the squeezed comb pairs is then demonstrated by evaluating the squeezing and anti-squeezing of the quadrature fields (position and momentum) through homodyne detection. Finally, the authors provide a qumode spectroscopy measurement to characterize the equidistance of the frequency modes spread over the quantum microcomb.

The realization of deterministic squeezed quantum microcombs can have significant impact on specific sectors of applied quantum science and technology, more precisely, spectroscopy, quantum metrology, and scalable quantum information processing based on the use of squeezed modes of light and homodyne detection techniques. The topic of the manuscript is thus quite timely. The length of the manuscript, presentation, and analysis all seem fitting. However, I think it is important that the authors revise their manuscript to address the following points:

As the authors are targeting a journal with a general audience, a brief discussion defining and contrasting probabilistic versus deterministic regimes of operation would be very useful for the introduction.

The authors mention in the introduction that the microcombs' large FSRs are the key to accessing individual modes – this statement can be presented more concretely, i.e., the FSRs are in the 10s of GHz accessible to off-the-shelf filters (as opposed to the finer FSRs of fiber- or bulk- resonator based combs).

The authors state that "The photon coincidence rate suffers from exponential decrease with the increase of photon number in a quantum state[18]". While accurate, this sentence is misleading when combined with ref. [18], where this issue is addressed via the use of qudits. The authors should rephrase accordingly.

When the authors refer to probabilistic approaches, they state that "no high resolution method is reported to verify the frequency equidistance". The authors should clarify why such a verification would be significant (especially since current approaches rely on energy conservation phenomena in resonators that can have their transmission spectra verified classically). More significantly, the authors should contextualize their spectroscopy work against other continuous-variable approaches.

Generally, while the current presentation of the state-of-the-art is relevant with its focus on other microcomb-based technologies, expanded context on other continuous-variable approaches is missing. The lack of this contrast makes the impact of the manuscript unclear. Moreover, while the results presented are very nice (especially considering that the setup is comprised of telecom-ready components), they are not contextualized against the state-of-the-art. For example, how do the squeezing metrics achieved contrast against those of the literature and the minimum values necessary for applications? (Both integrated and not). I also call the authors' attention to a thorough recent manuscript, arXiv:2103.10517, they might find relevant.

The authors should specify why their particular pump power was chosen.

The use of an EOM comb for the analysis is very nice – can the authors comment on the impact of RF noise on their measurement and on the potential scalability of the setup? The authors state the LO and pump are coherent with one another as they originate from the same source –the coherence is also due to the fact

that the EOM employs a coherent scattering operation (this is stated in the methods but not the main text).

For Figure 2, time is perhaps not the most useful choice of x-axis. I would suggest the authors use the phase-shifter phase as the x-axis (or as a second x-axis).

I do not believe Figure 3 clarifies the intense dips in anti-squeezing at modes 17 and 18. The authors should suggest why these occur. Are the intensities of the generated EOM comb equalized? If not, what impact do its variations have on the experiment?

The authors suggest that the modes nearer to the pump may become accessible in the future by using higher Q-factors and reducing the parametric oscillator threshold. If the issue is ASE noise from the EDFA, could a higher rejection bandpass filter not fix the problem? ASE is broadband – would it not be an issue for more modes studied here?

For Figure 4, the authors adjusted the escape efficiency to 77%. How was this adjusted? Was a new device employed for these measurements? Stability issues for higher efficiencies are briefly mentioned – the authors should go in depth on the nature and source of these issues (as it seems the setup can only run for short periods of time but this is not clear), as it is an important consideration for the platform.

The authors should clarify in their conclusion whether reducing the FSR is a potential way to scale the number of qumodes and how the ">1000" estimate was determined.

The manuscript suffers from some grammatical issues throughout (but nowhere to the point where the intended meaning is obfuscated). The authors should address this.

The methods and extended data for the manuscript feel very sparse and do not provide sufficient information to replicate the experiment. Some general remarks:

- The model numbers and key performance parameters of the equipment used should be stated (e.g., the resolution and model of the waveshaper)
- I could not find the driving frequency of the EOM comb used in Figure 2
- Does the servo control the CW wavelength at the source?
- The authors should provide details regarding their analysis methods, e.g., how the signal detected by the ESA is processed into the points plotted in Figs 2-4.

In conclusion, the manuscript does not meet, in this version, the publication criteria of Nature Communications. I would suggest the authors implement the revisions stated above, especially as the impact of the work is currently unclear, as a comparison against the current state-of-the-art is missing.

Reviewer #2 (Remarks to the Author):

Z. Yang et al. report the measurement of two-mode squeezed vacuum across many pairs of symmetrically placed modes around the pump, generated in a silica wedge resonator excited by a tapered fiber. Near the parametric oscillation threshold, such modes have been predicted to generate two-mode squeezed light. This was specifically studied for Kerr parametric oscillators by Y. Chembo in the theory paper, "Quantum dynamics of Kerr optical frequency combs below and above threshold: Spontaneous four-wave mixing, entanglement, and squeezed states of light," *Phys. Rev. A* 93, 033820 (2016)]. The current paper represents an experimental demonstration of this prediction across 20 pairs of modes. The experimental techniques are thoroughly developed and present convincing evidence for the existence of up to 1.6 dB of squeezing spread across pairs of these modes. I especially find the data in Fig. 4 to be commendable and novel. Overall, this work fits well within the recent surge in continuous variable quantum information protocols and their experimental realizations, both from academic and industry groups. Hence, I believe it will be of interest to the readership of Nature Communications. I do have several concerns regarding the introduction and the experimental details.

My major concern regarding the paper relates to the introduction. For example, the paper states, "Above the Kerr parametric oscillation threshold, microcombs behave classically ...". This is incorrect, as predicted theoretically by the work of Chembo in PRA 93 033820 (2016), and demonstrated experimentally both in second order bulk OPOs and third order integrated OPOs. In second order OPOs, the nonclassicality was demonstrated in experiments by one of the authors (e.g. Pfister group, PRA 74, 041804(R) 2006) as well as by one of the early experiments generating squeezed light (Heidmann et al. PRL 1987). In third order integrated OPOs, the nonclassicality has been demonstrated above threshold (Phys. Rev. Applied 3, 044005 (2015)). Hence, it is incorrect to say that above the Kerr parametric oscillation threshold, microcombs behave classically.

Moreover, the introduction misses several very relevant references that have shown multiple frequency quantum correlated beams using four-wave mixing. It also misses very relevant references to multi-frequency squeezed light generation in microresonators. A non-exhaustive list is:
Phys. Rev. Lett. 113, 023602 (2014), "Experimental Generation of Multiple Quantum Correlated Beams from Hot Rubidium Vapor"
Phys. Rev. Applied 3, 044005 (2015), "On-chip optical squeezing"
"Quantum Light from a Whispering-Gallery-Mode Disk Resonator," Phys. Rev. Lett. 106, 113901 (2011).
Refs 26 and 27 in the paper report single-mode squeezing generated from the same platforms as the last two references above, but these references above reported two-mode squeezing, which is more relevant to the current paper's results.

On the technical side, I have questions regarding the influence of Raman and Brillouin scattering on the measured squeezing levels. It is well-known that Raman and Brillouin scattering place severe restrictions in the squeezing level observed in silica fibers. See Bergman et al. Optics Letters 19, 290 (1994) (and other papers by the same authors), as well as Dong et al. Optics Letters 33, 116 (2008), "Experimental evidence for Raman-induced limits to efficient squeezing in optical fibers". The silica wedge resonator could be expected to suffer from these effects which would limit squeezing at the high threshold powers on chip. In fact, some of the authors have previously shown Brillouin lasers using the same platform, and hence one can reasonably expect Brillouin scattering (GAWBS or otherwise) to affect squeezing results. More recently, on-chip microresonators have shown effects of Brillouin noise on squeezing (see APL Photonics 5, 101303 (2020)). Could the authors provide a detailed discussion if they performed measurements to rule out Brillouin and Raman noise effects? This could be presented as a supplementary material or an extended methods section if needed.

Fig. 2(d) mentions "entanglement check" but we do not see any entanglement measurements presented in the paper. For that, it would be necessary to independently tune the two LO phases to measure $x_n + x_{-n}$ as well as $p_n + p_{-n}$ quadratures. The current way of scanning the phase of the EOM comb does not automatically produce this capability.

In squeezing demonstrations, especially on a new platform, it has become customary to report the laser source and balanced detectors used. Hence, these pieces of information should be included to enhance the reproducibility of the paper. The laser used does not go through a mode cleaning cavity, so it could have noise that can affect the squeezing levels measured if the CMRR of the balanced detection system is not sufficient.

What is the dependence of the squeezing level on the input pump power? For example, see Fig. 3 of the paper by Vaidya et al. (Ref. 29)).

REVIEWER COMMENTS

Reviewer #1 (Remarks to the Author):

Review of the manuscript “A squeezed quantum microcomb on a chip”

In the manuscript titled “A squeezed quantum microcomb on a chip” by Z. Yang et al., the authors demonstrate the realization of a deterministic quantum microcomb using an integrated silica microresonator. They use the microcomb as a platform for the generation of 40 continuous-variable (CV) quantum modes (qumodes) analyzed as 20 two-mode squeezed resonance pairs, emitted as signal and idler signals (generated through four-wave mixing within the micro resonator) spanning a 1 THz optical band at telecommunication wavelengths. The unconditional Einstein-Podolsky-Rosen (EPR) entanglement of the squeezed comb pairs is then demonstrated by evaluating the squeezing and anti-squeezing of the quadrature fields (position and momentum) through homodyne detection. Finally, the authors provide a qumode spectroscopy measurement to characterize the equidistance of the frequency modes spread over the quantum microcomb.

The realization of deterministic squeezed quantum microcombs can have significant impact on specific sectors of applied quantum science and technology, more precisely, spectroscopy, quantum metrology, and scalable quantum information processing based on the use of squeezed modes of light and homodyne detection techniques. The topic of the manuscript is thus quite timely. The length of the manuscript, presentation, and analysis all seem fitting. However, I think it is important that the authors revise their manuscript to address the following points:

- 1. As the authors are targeting a journal with a general audience, a brief discussion defining and contrasting probabilistic versus deterministic regimes of operation would be very useful for the introduction.*

Reply: We thank the reviewer for this helpful comment. We have added some clarifications regarding probabilistic versus deterministic quantum regimes in the introduction in the revised manuscript.

In the second paragraph, we added (changes highlighted in Italic): “Quantum architectures built upon probabilistic quantum states are not scalable without quantum memory, *which allows repeat-until-success strategies*. A quantum microcomb in the deterministic regime, *where the entanglement among different frequency modes can be deterministically generated and detected*, will be a significant step forward towards the scalable quantum architecture on photonic chips”.

In the third paragraph: “*The unconditional entanglement created by two-mode squeezing is between continuous optical fields, which can serve as quantum modes (qumodes) to encode quantum information through continuous-variable-based (CV) approaches for applications in universal quantum computing, ... Unlike probabilistic photonic qubit approaches, the*

unconditional entanglement in CV approaches enable the number of entangled quantum modes (qumodes) in a quantum state to be deterministically scaled up through frequency, time, or spatial multiplexing...

2. *The authors mention in the introduction that the microcombs' large FSRs are the key to accessing individual modes – this statement can be presented more concretely, i.e., the FSRs are in the 10s of GHz accessible to off-the-shelf filters (as opposed to the finer FSRs of fiber- or bulk- resonator based combs).*

Reply: We have added this statement in the revised manuscript: “*Access to individual quantum channels is possible through off-the-shelf wavelength-division-multiplexing filters thanks to microcombs' large free-spectral-ranges (FSRs), which range from a few GHz to 1 THz as opposed to the finer FSRs of fiber or bulk resonator-based combs.*”.

3. *The authors state that “The photon coincidence rate suffers from exponential decrease with the increase of photon number in a quantum state [18]”. While accurate, this sentence is misleading when combined with ref. [18], where this issue is addressed via the use of qudits. The authors should rephrase accordingly.*

Reply: We have followed the reviewer’s suggestion and removed reference [18] in this sentence to avoid any confusion to the audience. We would like to note that the use of qudits in [18] improves the scalability polynomially (not exponentially) by increasing d (Hilbert space size as d^n), and it cannot fully compensate for the exponential decrease of photon coincident rate with the increase of photon number n .

4. *When the authors refer to probabilistic approaches, they state that “no high resolution method is reported to verify the frequency equidistance”. The authors should clarify why such a verification would be significant (especially since current approaches rely on energy conservation phenomena in resonators that can have their transmission spectra verified classically). More significantly, the authors should contextualize their spectroscopy work against other continuous-variable approaches.*

Reply: For continuous-variable approaches, the frequency equidistance verification is important for homodyne detection, which requires local oscillators to be at the same frequency as the quantum modes. If the quantum microcomb has perfect frequency equidistance, then we only need a classical frequency comb as the local oscillators. On the other hand, perfect frequency equidistance is not necessary for photonic qubit-based quantum computing.

In the revised manuscript, we have added this clarification in the revised manuscript: “*The frequency equidistance of qumodes can simplify homodyne detection in CV quantum information processing, as a simple laser frequency comb can serve as local oscillators for*

all qumodes,” and we have removed “for probabilistic approach, no high resolution method is reported to verify the frequency equidistance”.

5. *Generally, while the current presentation of the state-of-the-art is relevant with its focus on other microcomb-based technologies, expanded context on other continuous-variable approaches is missing. The lack of this contrast makes the impact of the manuscript unclear. Moreover, while the results presented are very nice (especially considering that the setup is composed of telecom-ready components), they are not contextualized against the state-of-the-art. For example, how do the squeezing metrics achieve contrast against those of the literature and the minimum values necessary for applications? (Both integrated and not). I also call the authors’ attention to a thorough recent manuscript, arXiv:2103.10517, they might find relevant.*

Reply: (1) In the revised manuscript, we have introduced several milestone references in continuous-variable applications, including universal quantum computing [Physical Review Letters 82, 1784 (1999), Physical review letters 97, 110501 (2006), Physical review letters 112, 120504 (2014)], unconditional quantum teleportation [science 282, 706–709 (1998)], quantum dense coding [Physical review letters 88, 047904 (2002)], quantum secret sharing [Physical Review A 88, 042313 (2013)], and quantum key distribution [Nature 421, 238–241 (2003)]. We also referenced three existing ways to scale up the number of entanglement states: frequency, time and spatial multiplexing. We also added a sentence to introduce the common methods of generating squeezing includes bulk optical parametric oscillators and atomic vapor.

In the third paragraph, we have added: *“The unconditional entanglement created by two-mode squeezing is between continuous optical fields, which can serve as quantum modes (qumodes) to encode quantum information through continuous-variable-based (CV) approaches for applications in universal quantum computing [27,28,31], unconditional quantum teleportation [32], quantum dense coding [33], quantum secret sharing [34], quantum key distribution [35]. Unlike probabilistic photonic qubit approaches, the unconditional entanglement in CV approaches enables the number of entangled quantum modes (qumodes) in a quantum state to be deterministically scaled up through frequency [20,21,36], time [37-39], or spatial multiplexing [40], which provides a scalable physical platform for continuous-variable quantum computing. Squeezing and CV states are conventionally generated in bulk optical systems, such as optical parametric oscillators (OPOs) [23,44], and atomic vapor [45,46]...”*

(2) We have added discussion of the squeezing metrics reported in other literature (both integrated and free space) after we introduce our 1.6 dB squeezing result: *“Our 1.6 dB raw squeezing is among the highest raw squeezing measured for miniaturized Kerr optical parametric oscillators (OPOs) [56]. The highest squeezing ever achieved is 15 dB in a bulk $\chi^{(2)}$ OPO [61]. 6 dB single-mode squeezing was reported earlier in an integrated waveguide*

[53], which indicates that high squeezing is possible in integrated photonic systems. Recent theoretical studies have suggested that quantum error correction and fault-tolerant quantum computing is possible in photonic CV-based approaches [PRL, 112, 120504 (2014)] when squeezing reaches 10 dB [PRX, 8, 021054 (2018)]. We have also cited the recent manuscript arXiv:2103.10517 when we first introduce quantum microcomb in the first paragraph of the manuscript.

6. The authors should specify why their particular pump power was chosen.

Reply: In the current experimental condition, the amount of squeezing is primarily limited by the optical losses, and the specific pump power is not very critical to the amount of squeezing. We have included measurement of squeezing and anti-squeezing versus pump power for qumode (-4,4) in our revised manuscript. The amount of squeezing increases very little when the pump power is above ~50% of the parametric threshold. A similar level of squeezing can be achieved for a range of pump power.

We included the squeezing versus pump power measurement in Fig. 6(c) in the Methods section.

We have added this information in the revised manuscript: “The pump power in this experiment is set to 120 mW, which is 0.5 dB below the parametric threshold of 135 mW. The dependence of squeezing on the input pump power is discussed in the Methods section”.

In the Methods section, we included: “Dependence of squeezing on optical pump power. The dependence of squeezing and anti-squeezing on optical pump power is measured for qumode (-4,4), and is presented in Fig. 6(c). Ideally, when there is no optical loss, vacuum squeezing should increase with the pump power until the pump power reaches the OPO threshold. However, as the amount of squeezing in our experiment is primarily limited by optical losses,

the increase of squeezing can no longer be observed when the pump power is roughly above half of the OPO threshold. On the other hand, the anti-squeezing increases with the pump power. This observation is consistent with measurements in previous reports [55].”

7. *The use of an EOM comb for the analysis is very nice – can the authors comment on the impact of RF noise on their measurement and on the potential scalability of the setup? The authors state the LO and pump are coherent with one another as they originate from the same source –the coherence is also due to the fact that the EOM employs a coherent scattering operation (this is stated in the methods but not the main text).*

Reply: We thank the reviewer for this very insightful question. We have moved the sentence of EOM coherence to the main text in page 3: “The LOs and the resonator pump laser are coherent with each other since they are derived from the same cw-laser, *and the electro-optic modulators coherently transfer photons from the pump to the modulation sidebands.*”

The phase noise of the signal generator that drives the EOM comb will contribute to the phase fluctuation of the local oscillators. In our experiment, we have used a high-end commercial signal generator (Keysight, PSG E8257D) to drive the EOM comb, which has excellent phase noise performance. The root mean square (RMS) of phase jitter can be calculated from the standard single-sideband (SSB) phase noise on the datasheet, by integrating the phase noise from 100 kHz, the electrical spectrum analyzer (ESA) resolution bandwidth (RBW) we used in the squeezing measurement, to 400 MHz, the bandwidth of our balanced photodetection circuits (bandwidth’s theoretical upper limit). The RMS of phase jitter is calculated to be 1.59×10^{-3} rad (0.09°) for comb mode 1 (~ 22 GHz), and 0.0366 rad (2.1°) for comb mode 23 (~ 500 GHz). This level of phase fluctuation will not affect squeezing measurement at 1-2 dB level. For example, in “Optics Express, 14, 6930 (2006)”, 5.6 dB squeezing is measured while the phase fluctuation of the local oscillator is 4.3° . In our current experimental condition, if we assume 2 dB squeezing at mode 4, and 1 dB squeezing at mode 23 after optical loss, our calculation shows that the phase fluctuation will cause the measured squeezing to be 0.01 dB and 0.1 dB lower than the actual squeezing at mode 4 and mode 23, respectively.

In terms of scalability, the spectrum of the EOM comb can be broadened by using supercontinuum generation in highly nonlinear fiber (HNLF). A few thousand comb lines in an EOM comb have been demonstrated. It should be noted that the N -th comb line in the EOM comb has N times the RMS phase jitter of the 1st comb line in the EOM comb. Therefore, when scaling up the number of comb lines in an EOM comb through supercontinuum generation for squeezing measurement, the phase noise of the signal generator should be improved accordingly to maintain the low phase fluctuation of the local oscillators. A possible way to obtain exceptional phase noise performance for the EOM comb is to stabilize the EOM comb to stable optical references. This has been demonstrated by one

of the co-authors in an earlier paper through electro-optical frequency division (Science, 345, 309 (2014)).

We have added a paragraph in the method section regarding the impact of EOM phase noise and the scalability:

“The phase noise of the signal generator that drives the EOM comb contributes to the phase fluctuation of the local oscillator, which could potentially affect squeezing measurement [77]. The root mean square (RMS) of phase jitter from the signal generator can be calculated from its single-sideband (SSB) phase noise (signal generator model: Keysight, PSG E8257D) by integrating the phase noise from the electrical spectrum analyzer (ESA) resolution bandwidth (RBW) used in the squeezing measurement (100 kHz), to the bandwidth of our balanced photodetection circuit (400 MHz). The RMS of phase jitter is calculated to be 1.59×10^{-3} rad (0.09°) for comb mode 1 (~ 22 GHz), and is 0.0366 rad (2.1°) for comb mode 23 (~ 0.5 THz). At the current experimental condition, assuming 2 dB squeezing at mode 4, and 1 dB squeezing at mode 23 after optical losses, the phase fluctuation will cause the measured squeezing to be 0.01 dB and 0.1 dB lower than the actual squeezing at mode 4 and mode 23, respectively. It should be noted that the N -th comb line in the EOM comb has N times the RMS phase jitter of the 1st comb line in the EOM comb. Therefore, when scaling up the number of comb lines in an EOM comb through supercontinuum generation [60] for squeezing measurement, the phase noise of the signal generator should be improved accordingly to maintain the low phase fluctuation of the local oscillators. A possible way to obtain exceptional phase noise performance for the EOM comb is through electro-optical frequency division, where the signal generator is synchronized to stable optical references [79]”.

8. *For Figure 2, time is perhaps not the most useful choice of x-axis. I would suggest the authors use the phase-shifter phase as the x-axis (or as a second x-axis).*

Reply: We have added a panel in Figure 2(b) to illustrate the voltage we applied to the phase modulator for ramping the phase of the local oscillator. We've also included a measured V_π of 2.3 Volt for the phase modulator in the revised manuscript. As the voltage ramping range is from -4.88 to 4.88 Volt, the phase ramping range will be from -2.1π to 2.1π . These are all included in Figure 2(b) in the revised manuscript.

9. *I do not believe Figure 3 clarifies the intense dips in anti-squeezing at modes 17 and 18. The authors should suggest why these occur. Are the intensities of the generated EOM comb equalized? If not, what impact do its variations have on the experiment?*

Reply: (1) The sudden change in squeezing/anti-squeezing in certain frequency modes (10, 17, 18, etc) is most likely a result of the spatial-mode interaction (mode-crossing) in the microresonator. Spatial-mode interaction happens when the frequencies of two transverse mode families are very close, and there are linear coupling among them (e.g., scattering)

[PRL 100, 103905 (2008), PRL 113, 123901 (2014)]. The spatial-mode interaction can reduce the amount of squeezing/anti-squeezing because it changes resonator mode dispersion at the interaction wavelengths, and it introduces a dissipative (loss) channel/mode for the optical fields.

The mode spectrum measurement in Fig.3(b) indicates that: (a) an avoided mode crossing at mode -8 and -9 and (b) above mode 19, and below mode -18, the mode frequencies deviate from the ideal 2nd order dispersion. These are typical evidence of spatial-mode-interaction. The positions of the spatial-mode interaction in the mode spectrum (-8, -9, < -18, and > 19) coarsely align with the anti-squeezing dips at mode 10, and from mode 17 to 23. Although this is not direct proof that anti-squeezing dips are caused by spatial-mode interaction, it nevertheless presents the possible correlation between them. A more systematic study is required to fully understand the physics mechanism between spatial-mode-interaction and squeezing/anti-squeezing, and we will study this in detail in a follow-up manuscript in the future.

In the revised manuscript, we have included a few sentences of how the spatial-mode interaction can affect the squeezing/anti-squeezing, and added two related references:

“We suspect this is caused by spatial-mode interaction between different transverse mode families in the resonator, which not only modifies local dispersion [63], but provides a path to dissipate optical fields from the squeeze-generating mode to another spatial mode [64]. The spatial-mode-interaction can be identified by measuring the frequency spectrum of a resonator..... The mode numbers that are affected by spatial-mode-interaction in the mode spectrum coarsely align with that of the dips in anti-squeezing measurement. More systematic studies will be performed in the future to understand the mechanism of how spatial-mode-interaction affects squeezing and anti-squeezing. Finally, the impact of spatial-mode interaction can be eliminated in the future by using a microresonator with a single transverse mode family.”

(2) The intensities of local oscillator pairs generated from the EOM comb are equalized by adjusting the attenuation level in the waveshaper. Our waveshaper has 0.1 dB attenuation resolution for each frequency channel, and thus allows us to fine adjust intensities of local oscillator pairs. If the intensities are not equalized, the homodyne detection will pick up individual quadrature, which will be directly added to the entire noise level in the homodyne detection. This will cause the measured value of anti-squeezing to be higher than the actual value, and the measured value of squeezing to be smaller than the actual value. Therefore, this is unlikely to be the cause of the dips in anti-squeezing at modes 17 and 18.

10. *The authors suggest that the modes nearer to the pump may become accessible in the future by using higher Q-factors and reducing the parametric oscillator threshold. If the issue is ASE noise from the EDFA, could a higher rejection bandpass filter not fix the problem? ASE is broadband – would it not be an issue for more modes studied here?*

Reply: The reviewer is correct. Ideally, the problem can be fixed by using a high rejection bandpass filter with its bandwidth smaller than the resonator FSR. In our experiment, we have already used a fiber-Bragg grating filter to filter out most of the ASE noise from EDFA. The FWHM of this filter is about 44 GHz, and the full-width 20 dB rejection bandwidth is 59 GHz. Therefore, some ASE noise near the pump wavelength can still pass the filter. For our 22 GHz FSR resonator, the first pair of modes near the pump laser will be affected by the ASE noise. The second and third pairs of modes are on the edge of the high rejection bandwidth, and we excluded them in our manuscript for precaution. Measurements for qumodes (-4,4) to (-23,23) are not affected by the ASE noise in our current setup.

We have started looking for alternative filters with narrower bandwidth for our setup a few months ago. So far, the filters we tested all have very large photothermal effect and cannot be used for the experiment. We still have a few candidate filters to arrive later this year, and we also plan to build our own temperature-controlled filter to address this issue in the future.

We have added a few sentences to include the information of the filter we use, and also point out that the ASE issue can be fixed by using a narrow band, high rejection bandpass filter: On page 2: *“At the through port of the resonator, a narrow-band fiber Bragg grating (FBG) filter is used to filter out the pump field and the amplified spontaneous emission (ASE) noise from the fiber amplifier.”*

On page 4: *“This can be addressed in the future by using a filter with bandwidth much smaller than the FSR of the resonator, or by increasing the intrinsic quality factor of the cavity and reducing the parametric oscillation threshold to eliminate the need for the EDFA.”*

11. *For Figure 4, the authors adjusted the escape efficiency to 77%. How was this adjusted? Was a new device employed for these measurements? Stability issues for higher efficiencies are briefly mentioned – the authors should go in depth on the nature and source of these issues (as it seems the setup can only run for short periods of time but this is not clear), as it is an important consideration for the platform.*

Reply: The escape efficiency is adjusted by tuning the relative position between the tapered fiber and the optical microresonator. In our experiment, the microresonator sits on a nanopositioning piezo stage, where the microresonator’s relative position to the tapered fiber can be adjusted by tuning the voltage on the piezo. To reduce the escape efficiency, we will move the tapered fiber slightly away from the optical modes.

We have added a sentence in the revised manuscript:

“In this experiment, the escape efficiency is adjusted by varying the relative position between the microresonator and the tapered fiber [PRL, 85, 74 (2000)]”.

In terms of the stability of our setup: the relative position between the tapered fiber and the microresonator might drift slightly due to the thermal expansion of the setup and the optical table. Typically, the coupling between the tapered fiber and microresonator is stable for

several hours at any escape efficiency, which is sufficient to finish squeezing measurement in figure 2 and 3. The stability issue is only significant for the qumode spectroscopy measurement in figure 4, as this measurement takes up to 18 hours and the temperature in the lab can change > 1 degree C during this period. By testing the coupling repeatedly, we found that the relative position between the tapered fiber and this specific microresonator is most stable at a position with 77% escape efficiency.

There is a clear path for the microresonator system to fully address the stability issue: the stability can be greatly improved in the future by (1) packaging the tapered fiber and the microresonator together, or (2) integrating the coupling waveguide and the microresonator onto the same chip.

We have added this in the revised manuscript:

“The stability of the escape efficiency can be dramatically improved by packaging the microresonator systems [Nat. Photon 13, 25 (2019)], or by integrating the coupling waveguide and the resonator on the same chip [Nat. Photon 12, 297 (2018)]”.

12. *The authors should clarify in their conclusion whether reducing the FSR is a potential way to scale the number of qumodes and how the “>1000” estimate was determined.*

Reply: In the > 1000 modes estimation, we calculated squeezing with microresonator parameters of 10 GHz FSR, dispersion $D_2/2\pi = 10$ kHz, loaded cavity linewidth of 1 GHz, 90% escape efficiency, and the pump power is 1 dB below the parametric oscillation threshold. In our calculation, we see that more than 500 pairs of qumodes with obvious two-mode squeezing (see figure below). This set of resonator parameters is readily available in SiN microresonator that is used to generate soliton microcombs [Nature Photonics 14, 486-491 (2020)].

In the revised manuscript, we rephrased this part to emphasize less on the specific number of modes, but more on the approaches that can increase the optical bandwidth of Kerr parametric gain and quantum microcomb.

On page 2: we replaced the “1000 qumodes” sentence with: *“The optical span of a quantum microcomb is ultimately set by the chromatic dispersion, which determines the bandwidth of Kerr parametric gain.”*

In the discussion section, we added: *“The optical span of quantum microcombs will ultimately be limited by the microresonator dispersion, which sets the bandwidth of Kerr parametric gain. Through dispersion engineering, Kerr parametric sidebands that are $\sim \pm 80$ THz away from the pump frequency have been reported in microresonators [Nat. Photon 13, 701–706 (2019)], which indicates the possibility of creating hundreds or thousands of qumodes in a single microresonator.”*

Figure R1: Numerical calculation of squeezing and anti-squeezing versus cavity mode number for a SiN microresonator with: 10 GHz FSR, dispersion $D_2/2\pi = 10$ kHz, loaded cavity linewidth of 1 GHz, 90% escape efficiency, pump power that is 1 dB below the parametric oscillation threshold, and no optical loss.

13. The manuscript suffers from some grammatical issues throughout (but nowhere to the point where the intended meaning is obfuscated). The authors should address this.

Reply: We thank the reviewer for this helpful comment. We have checked the grammar carefully in the revised manuscript.

14. The methods and extended data for the manuscript feel very sparse and do not provide sufficient information to replicate the experiment. Some general remarks:

- The model numbers and key performance parameters of the equipment used should be stated (e.g., the resolution and model of the waveshaper).

Reply: We have included the vendors, model numbers and some key parameters of the (a) pump laser, (b) signal generator in the EOM comb, (c) waveshaper, and (d) photodiodes in the revised manuscript. We have also added more experimental details in the Methods section:

Pump laser: New Focus External Cavity Diode Laser, Model: TLB-6700;

Signal generator: Keysight E8257D PSG.

Waveshaper: Finisar 1000 A. Filter bandwidth setting resolution: ± 5 GHz.

Photodiodes in balanced homodyne system: JDSU, ETX 300 T. Common-mode rejection ratio: > 31 dB.

15. I could not find the driving frequency of the EOM comb used in Figure 2.

Reply: In the revised manuscript, we have included the driving frequency of the EOM comb in the method section.

16. *Does the servo control the CW wavelength at the source?*

Reply: Yes, the output of the servo is connected to the frequency modulation port of the laser to control the frequency of the cw laser. We have clarified this in the revised manuscript.

17. *The authors should provide details regarding their analysis methods, e.g., how the signal detected by the ESA is processed into the points plotted in Figs 2-4.*

Reply: For the quadrature noise variance vs time measurement (Fig. 2 and Fig. 4(b)), the only signal processing is a 30-point moving average of the original ESA traces to smooth out fast fluctuations. This information is provided in the original manuscript. The squeezing and anti-squeezing values in Fig. 3 and Fig. 4 are obtained by averaging the displayed minima/maxima in each quadrature noise variance vs time figure. The error bars are concluded with a 95% confidence interval under t-distribution. The qumode spectrum values in Fig. 4 (d) are achieved by fitting Fig. 4 (c) with Lorentzian lineshape and then finding the center. We have included these analysis methods in the manuscript.

Reviewer #2 (Remarks to the Author):

Z. Yang et al. report the measurement of two-mode squeezed vacuum across many pairs of symmetrically placed modes around the pump, generated in a silica wedge resonator excited by a tapered fiber. Near the parametric oscillation threshold, such modes have been predicted to generate two-mode squeezed light. This was specifically studied for Kerr parametric oscillators by Y. Chembo in the theory paper, "Quantum dynamics of Kerr optical frequency combs below and above threshold: Spontaneous four-wave mixing, entanglement, and squeezed states of light," *Phys. Rev. A* 93, 033820 (2016)]. The current paper represents an experimental demonstration of this prediction across 20 pairs of modes. The experimental techniques are thoroughly developed and present convincing evidence for the existence of up to 1.6 dB of squeezing spread across pairs of these modes. I especially find the data in Fig. 4 to be commendable and novel. Overall, this work fits well within the recent surge in continuous variable quantum information protocols and their experimental realizations, both from academic and industry groups. Hence, I believe it will be of interest to the readership of *Nature Communications*. I do have several concerns regarding the introduction and the experimental details.

1. My major concern regarding the paper relates to the introduction. For example, the paper states, "Above the Kerr parametric oscillation threshold, microcombs behave classically ...". This is incorrect, as predicted theoretically by the work of Chembo in *PRA* 93 033820 (2016), and demonstrated experimentally both in second order bulk OPOs and third order integrated OPOs. In second order OPOs, the nonclassicality was demonstrated in experiments by one of the authors (e.g. Pfister group, *PRA* 74, 041804(R) 2006) as well as by one of the early experiments generating squeezed light (Heidmann et al. *PRL* 1987). In third order integrated OPOs, the nonclassicality has been demonstrated above threshold (*Phys. Rev. Applied* 3, 044005 (2015)). Hence, it is incorrect to say that above the Kerr parametric oscillation threshold, microcombs behave classically.

Reply: The review is correct. Our sentence "microcombs behave classically above parametric threshold" is not accurate. We have removed this sentence in the revised manuscript.

2. Moreover, the introduction misses several very relevant references that have shown multiple frequency quantum correlated beams using four-wave mixing. It also misses very relevant references to multi-frequency squeezed light generation in microresonators. A non-exhaustive list is: *Phys. Rev. Lett.* 113, 023602 (2014), "Experimental Generation of Multiple Quantum Correlated Beams from Hot Rubidium Vapor", *Phys. Rev. Applied* 3, 044005 (2015), "On-chip optical squeezing". "Quantum Light from a Whispering-Gallery-Mode Disk Resonator," *Phys. Rev. Lett.* 106, 113901 (2011). Refs 26 and 27 in the paper report single-mode squeezing generated from the same platforms as the last two references above, but these references above reported two-mode squeezing, which is more relevant to the current paper's results.

Reply: We have added these references in the revised manuscript, and introduced squeezing generation from atomic vapor in the introduction.

3. *On the technical side, I have questions regarding the influence of Raman and Brillouin scattering on the measured squeezing levels. It is well-known that Raman and Brillouin scattering place severe restrictions in the squeezing level observed in silica fibers. See Bergman et al. Optics Letters 19, 290 (1994) (and other papers by the same authors), as well as Dong et al. Optics Letters 33, 116 (2008), "Experimental evidence for Raman-induced limits to efficient squeezing in optical fibers". The silica wedge resonator could be expected to suffer from these effects which would limit squeezing at the high threshold powers on chip. In fact, some of the authors have previously shown Brillouin lasers using the same platform, and hence one can reasonably expect Brillouin scattering (GAWBS or otherwise) to affect squeezing results. More recently, on-chip microresonators have shown effects of Brillouin noise on squeezing (see APL Photonics 5, 101303 (2020)). Could the authors provide a detailed discussion if they performed measurements to rule out Brillouin and Raman noise effects? This could be presented as a supplementary material or an extended methods section if needed.*

Reply: (1) The Brillouin shift in silica is ~ 10.9 GHz at 1550 nm, and the typical full width half maximum (FWHM) bandwidth of Brillouin gain is 20 - 60 MHz. Therefore, Brillouin scattering can be only observed in the microresonator when the free-spectral-range (FSR) of the microresonator matches the Brillouin shift precisely [*Nat. Photon* 6, 369–373 (2012)]. In our experiment, we have specifically chosen 22 GHz to be our resonator FSR, which is completely out of the Brillouin gain spectrum and there is no Brillouin scattering in the resonance modes in this microresonator. This is a critical difference between the fiber system and microresonator system, as the small microresonator can have FSR much larger than the Brillouin shift frequency, and thus do not suffer from the Brillouin scattering effect.

We want to note that in APL Photonics 5, 101303 (2020), the authors generated bright squeezing from Sagnac interferometer, which is fundamentally different from our experiment. Also in their figure 4, they predicted squeezing > 10 dB in their system through theoretical calculation, which suggests that Brillouin and Raman are not the limiting factors in their squeezing measurement.

(2) The Raman scattering generates spontaneous photons at the longer wavelength side of the pump laser, and thus could potentially limit the quantum correlation between signal and idler quumodes. In our current experiment, the quantum microcomb span is only ± 0.5 THz of the pump laser, while the Raman gain peak in silica is at ~ 13 THz. The Raman gain at 0.5 THz in silica is less than 2% of the peak Raman gain [Agrawal, Govind P. "Nonlinear fiber optics."], and the peak Raman gain is less than the Kerr parametric oscillation gain that generates the two-mode squeezing [PRL, 93 (8), 083904 (2004)]. Thus, the impact of Raman on squeezing is negligible in our current experiment. We have also calculated the amount of squeezing with and without Raman scattering term numerically (figure below), and no

difference of squeezing can be observed. In this calculation, we have assumed 83% cavity escape efficiency (same as our experiment condition) and perfect system efficiency (no optical loss).

This result does not contradict previous fiber experiments. In fiber experiments, the pump laser is usually pulsed (~ 100 s fs pulse width), and the peak optical power can reach kW level (*Optics Letters* 33, 116 (2008)). The propagation distance (fiber length) is typically a few meters to 10s of meters. However, our experiment is performed with continuous-wave laser and 100s mW power, and our resonator only has a diameter of 3 mm. Therefore, the Raman scattering strength in our resonator is much smaller than the previous fiber experiments.

We have added the following sentences in the method section in the revised manuscript:

“It should be noted that the Brillouin scattering does not affect the squeezing process in our resonator, as the resonator FSR is designed to be completely out of the Brillouin gain bandwidth [Nat. Photon 6, 369–373 (2012)]. Raman scattering in silica has its peak gain at 13 THz away from the pump, and the peak Raman gain is smaller than the Kerr parametric gain in microresonators with anomalous dispersion [PRL, 93 (8), 083904 (2004)]. As the optical span of the quantum microcomb is only ± 0.5 THz around the pump, the Raman gain within our microcomb span is only 2% of the peak Raman gain, and it has a negligible effect in our current experiment.

Figure R2: Calculation of squeezing level with and without Raman gain. The calculation uses our silica resonator parameters, 83% escape efficiency and assumes no optical loss.

4. Fig. 2(d) mentions "entanglement check" but we do not see any entanglement measurements presented in the paper. For that, it would be necessary to independently tune the two LO phases to measure $x_n + x_{-n}$ as well as $p_n + p_{-n}$ quadratures. The current way of scanning the phase of the EOM comb does not automatically produce this capability.

Reply: The review is correct. We have changed the “entanglement check” to “quantum correlation check” in the revised manuscript.

5. *In squeezing demonstrations, especially on a new platform, it has become customary to report the laser source and balanced detectors used. Hence, these pieces of information should be included to enhance the reproducibility of the paper. The laser used does not go through a mode cleaning cavity, so it could have noise that can affect the squeezing levels measured if the CMRR of the balanced detection system is not sufficient.*

Reply: In the revised manuscript, we have included the information of laser source (New Focus TLB-6700) and balanced detectors (InGaAs photodiodes JDSU, ETX 300T), and other vendor and model information.

The common-mode rejection ratio (CMRR) for our balanced detector is > 31 dB (Figure R3 (a)), which is measured by introducing a strong modulation peak to the laser. This is sufficient for our squeezing measurement as at our squeezing measurement condition, the laser technical noise is only 19 dB higher than the shot noise level (Figure R3 (b)).

Figure R3: (a) Common-mode rejection ratio (CMRR) measurement. To quantitatively measure the CMRR, the laser is intensity modulated to create a spike at 10 MHz. We then measure the electrical spectrum of the photodetector in the unbalanced (blue trace, measured by blocking light into one of the detectors), and the balanced situation (red trace). The modulation peak of the balanced case is observed to be ~ 31 dB lower than the unbalanced case. Thus, a CMRR of 31 dB can be concluded. (b) Characterization of unbalanced noise to ensure that the 31 dB CMRR is sufficient for the squeezing measurement. We measured the noise level of unbalanced (blue) and balanced (red) photodiode outputs versus time. The measurement is performed with ~ 17 mW optical incident power on the PDs, and the noise is measured at 2.7 MHz offset frequency with 100 kHz resolution bandwidth. The noise reaches the shot noise limit in the balanced case, which is 19 dB lower than the unbalanced noise. Therefore, we can conclude that the 31 dB CMRR is sufficient to measure a few dB squeezing in our experiment.

We have added CMRR information in the revised manuscript: “The electrical circuit for balancing the photodiodes is home-built [PRL, 112, 120505 (2014)], and a common-mode rejection ratio of 31 dB is measured.”

6. What is the dependence of the squeezing level on the input pump power? For example, see Fig. 3 of the paper by Vaidya et al. (Ref. 29)).

Reply: We did measure the dependence of the squeezing level of mode (-4,4) on the input power, which is shown in the figure below. We have normalized the input power to the parametric oscillation threshold in this figure. The squeezing first increases with pump power but stops increasing above certain input power level, while the anti-squeezing continuously increases with pump power. These trends are identical to Fig. 3 in the manuscript of Vaidya et al. We have included this Fig. 6(c) in the revised manuscript.

REVIEWERS' COMMENTS

Reviewer #1 (Remarks to the Author):

The authors thoroughly address all of the points we raised in the original review, both in the rebuttal and revised manuscript. We recommend the revised manuscript be accepted for publishing in Nature Communications.

Reviewer #2 (Remarks to the Author):

The manuscript has improved by a lot over the original version. In response to the comments of both reviewers, the authors have added important experimental details that were missing, and corrected erroneous technical statements. The detailed description of the relationship between modulation frequency f_m and the repetition rate f_r is also a good addition. A minor comment is regarding the Raman gain peak being 13 THz away from the pump. While this is not a problem for the current experiment spanning ± 0.5 THz, the future wideband combs the authors discuss for scalability could be hindered by this issue. I believe the current results stand on their own and hence would leave this to future studies.

Reviewer #1 (Remarks to the Author):

The authors thoroughly address all of the points we raised in the original review, both in the rebuttal and revised manuscript. We recommend the revised manuscript be accepted for publishing in Nature Communications.

Reviewer #2 (Remarks to the Author):

The manuscript has improved by a lot over the original version. In response to the comments of both reviewers, the authors have added important experimental details that were missing, and corrected erroneous technical statements. The detailed description of the relationship between modulation frequency f_m and the repetition rate f_r is also a good addition. A minor comment is regarding the Raman gain peak being 13 THz away from the pump. While this is not a problem for the current experiment spanning ± 0.5 THz, the future wideband combs the authors discuss for scalability could be hindered by this issue. I believe the current results stand on their own and hence would leave this to future studies.

Reply: We thank the reviewer for his/her comments. We have added a sentence following the discussion of Raman in the Method section: *“The effect of Raman scattering on wideband quantum microcombs will be studied in the future.”*